# Responsive Parametric Building Free Forms Determined by Their Elastically Transformed Steel Shell Roofs

**Aleksandra Prokopska and Jacek Abramczyk ***

Department of Architectural Design and Engineering Graphics, Rzeszow University of Technology,
Al. Powstańców Warszawy 12, 35-959 Rzeszów, Poland; aprok@prz.edu.pl
* Correspondence: jacabram@prz.edu.pl; Tel.: +48-795-486-426

**Abstract:** The article concerns the unconventional architectural forms of buildings roofed with transformed shells made up of thin-walled steel fold sheets, and a parametric description of how they are shaped. Complicated deformations of flanges and webs, as well as the complex static–strength work of the folds in a shell roof, demand the creation of simplified models regarding the parameterization of such shells and their integration with the general forms of the buildings. To obtain favorable results, it was necessary to write computer applications because of both the complicated problems related to the significant limitations of the transformations, as well as the great possibilities of shaping shell roofs by means of directrices of almost free shape and mutual position. The developed procedures enable the prediction of shapes and states of all the folds in the designed shell. They take account of two basic conditions related to these restrictions, which guarantee that the folds encounter little resistance when matching their transformed forms to the roof directrices, and that their initial effort was as low as possible. The developed procedures required solving a number of issues in the fields of architecture, civil engineering, and structures, and are illustrated with an example of shaping one unconventional architectural form. The interdisciplinary study explains a new insight into shaping such forms.

**Keywords:** corrugated shell roof; free-form building; architectural form; folded sheet; thin-walled profile; shape transformation; steel construction

## 1. Introduction

Curved shells, whether stiffened or not with structural ribs, that carry dead and live loads have been a great challenge for the engineers and architects of every era. In subsequent epochs, not only were the materials, weight, static diagrams, stiffness of structural elements and joints, spans, and durability of the designed shells and entire buildings changed, but their visual [1] attractiveness, form coherence, and architectural sensitivity to the natural and built environments have been modified as well [2].

Since the Roman times, single-curvature shell vaults have been used more and more often, including especially barrel and cross vaults. Since the Gothic style, doubly-curved roof shells with a positive Gaussian curvature [2] have been built, which was a result of the expected compressive stresses in them [3]. Stiffening or supporting ribs have been used to join complete smooth shells into a composite shell structure [3].

The issues related to the search for thin-walled concrete shells transferring a characteristic load were presented by H. Isler inter alia in [4]. He created models based on the nature-based solutions and conducted experiments with surfaces similar to the so-called minimum surfaces.

Examples and procedures preventing the destruction of reinforced concrete shells were presented by Foraboschi in [5]. An additional factor that causes damages to roof shells is dynamic influences.

Foraboschi discussed the appropriate procedures to prevent unfavorable dynamic influences in [5,6]. In addition, in the areas affected by seismic influences, the roof shell structure should be designed to improve the durability and bearing capacity of the designed building [7].

Shell roofs can simultaneously perform various functions. The multidimensionality of the issues related to their design, construction, and maintenance requires a comprehensive, parametric approach to shaping diversified unconventional architectural forms and the structures of entire buildings roofed with the shells [8]. The aspects of the parametric description of the architectural forms are under consideration in this paper. In particular, this paper proposes a parameterization of such roofs that is made up of many nominally plane thin-walled folded steel sheets connected to each other along their longitudinal edges into single continuous plane strips. Subsequently, each strip is transformed into a corrugated shell roof (Figure 1) as a result of spreading the strip on two skew directrices passing transversally to the fold's directions [1].

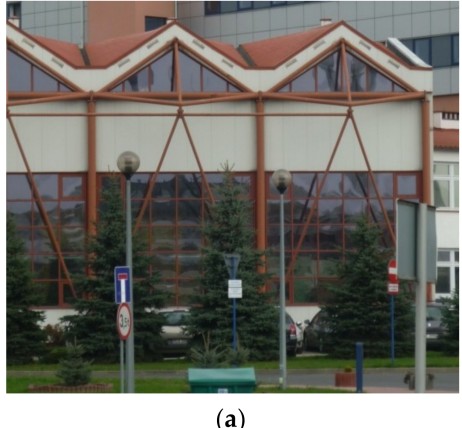
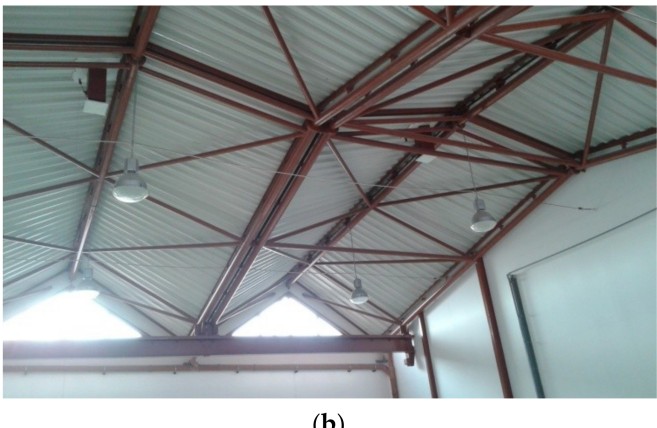

(a)                                                                                      (b)

**Figure 1.** A roof shell structure composed of many transformed fold shell strips: (**a**) view from the outside; (**b**) view from the inside.

One of the characteristics of the considered transformed shells is that the directrices stiffen their transverse edges, but their longitudinal edges must be stiffened with additional edge elements in order to maintain the straightness of the border folds in each strip (Figure 1b) [1]. This is the first limitation in shaping the transformed shells, which induces additional effort besides the initial stresses caused by the shape transformations that are determined by arranging and pressing each folded strip to the roof directrices. It should be noted that the technique and direction of the pressure of each fold to the directrices should result in the smallest cross-sectional change of the fold, so as not to unduly reduce its capacity and stability.

If the directrices are parallel to each other (Figure 2c) [9], the shape transformations do not result in significant values regarding the initial stresses, because the curvatures that are used in most building shell roofs and roof directrices are not unduly large, and the stresses need not be included in the static-strength calculations [1]. In this case, the shells can take the forms of various cylindrical surfaces. However, if the directrices are not parallel lines [8,9], the folds are twisted (Figure 2a) or twisted and bent (Figure 2b,d). Moreover, the deformations of their webs and flanges can be considerable and different both along the length of the same fold and in the adjacent folds in a shell. These differences may result in substantial values of compressive stresses in the fold's half-lengths and tensile stresses at both of the transverse ends of each twisted fold, depending on the degree of the fold's twist.

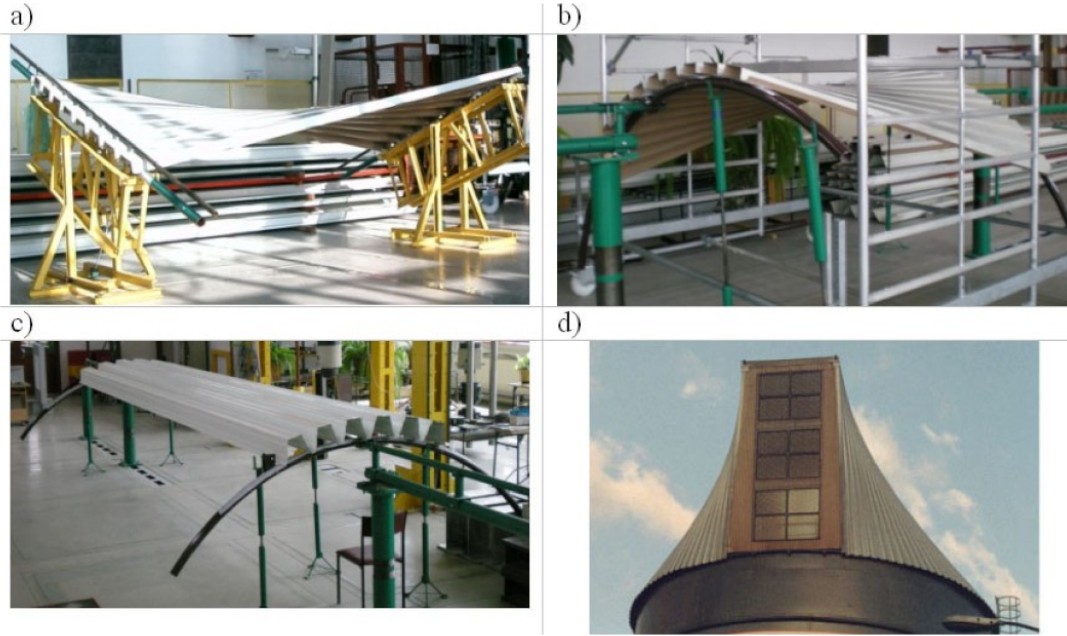

**Figure 2.** Corrugated shell sheets spread on: (**a**) straight directrices; (**b**,**c**) curved directrices; and (**d**) straight and curved directrices.

The longitudinal edges of each twisted or twisted and bent fold, similar to the longitudinal axes of each pair of adjacent folds in the shell, are skew straight lines, which result in different cross-sections of each shell fold along its length. The experimental tests and computer analyses carried out by Reichhart and Abramczyk [1,8] showed that each such transformed shell fold works effectively when its contraction occurs halfway along its length. In this case, the tensile stresses appearing at both transverse ends of the fold are comparable, and they balance the compressive stresses appearing in the middle part of the fold along the length.

Moreover, the distribution of the above-mentioned stresses in the fold's flanges and webs shows that each such transformed fold tends to bend its longitudinal edges with the convexity halfway along the length of the longitudinal edges directed to the outside, thereby affecting the adjacent folds in a shell (Figure 3). The action of the fold has to be balanced by the forces affecting the fold and coming from its neighboring folds. Transformed folds are designed to carry their own weight as well as the characteristic load, so the initial effort resulting from the shape transformations has to be limited appropriately.

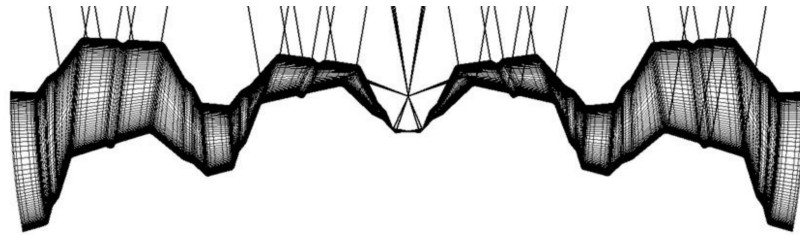

**Figure 3.** An exact computational folded model of a nominally plane folded sheet transformed into a shell shape.

These influences of adjacent folds in shells have not yet been researched well enough, and the descriptions presented in the available literature are too general. However, the results of the experimental tests and computer analyses [1,10] indicate a large variety of possible unconventional forms of thin-walled folded sheeting transformed from flat to spatial forms, despite these initial

stresses. The variety results from the great freedom in the adoption of the shapes and the mutual position of roof directrices, as well as the location of the fold directions in relation to the directions of directrices. The non-perpendicularity of the directions of folds and directrices results in an oblique cutting of both transverse ends of the shell folds [8]. A parametric description of the relationships between these supporting conditions of shell folds and the shapes of these folds allows the use of computer programming technology to create simplified smooth models of these folds and entire shells for engineering developments. The supporting conditions depend on the shapes and mutual position of the roof directrices, and are called boundary conditions. For scientific purposes, that is, for an incremental non-linear dynamic analysis of the static and strength properties of the shell folds as structural elements (Figure 4) [10], authors use advanced programs such as ADINA, for example [11].

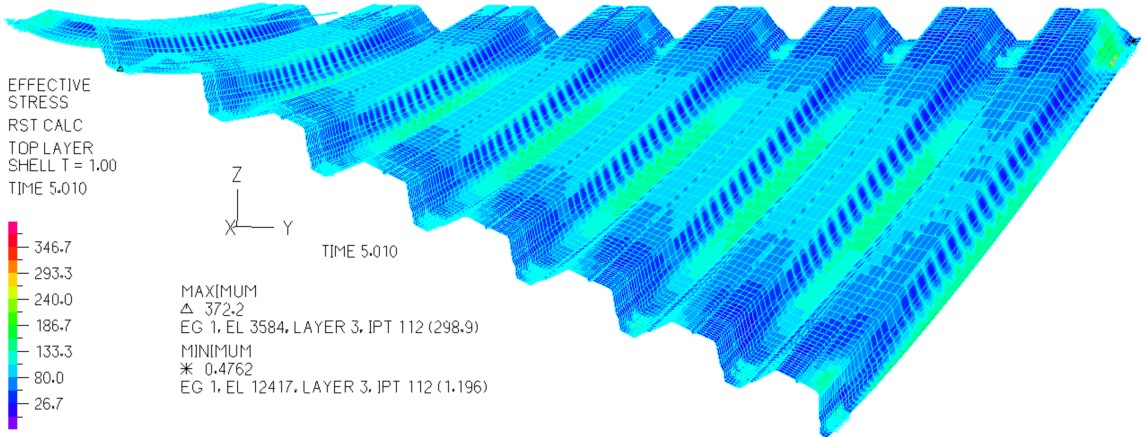

**Figure 4.** An exact computational folded model of a nominally plane folded sheet transformed into a shell shape.

In order to understand the parametric description presented in the present article, some problems should be explained. For engineering developments, regular geometrical surfaces (Figure 5) are employed to model subsequent shell folds in each transformed shell roof. It is possible to find only one shell shape of a transformed fold, which is assigned to the calculated border conditions resulting from the geometrical supporting conditions in the shell, such that its transformation is effective [1,8]. The characteristic of this shape is that its contraction passes halfway along its length transversally to the fold's longitudinal axes. Figure 5 shows the contraction line, which is called the line of striction, and is denoted as $s$. As a result, the effectively transformed fold can be spread on the roof directrices relatively freely, that is, with the lowest possible pressing forces. Furthermore, its impact on the forms of adjacent folds in the shell is the least possible. In this way, the effort is optimized to the lowest possible level. The above-mentioned pressing forces are needed to fix the fold's ends to the roof directrices.

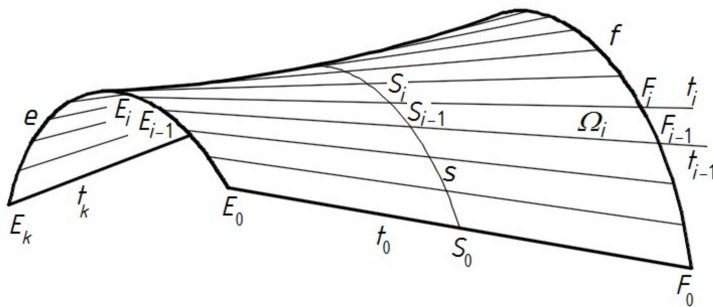

**Figure 5.** Simplified smooth shell model of a transformed shell roof with the $s$ line of striction and rulings $t_i$ modeling the longitudinal borders of subsequent folds in a shell.

For the effective fold transformations, interdependence between the geometrical supporting conditions and the obtained shell forms of a transformed fold can be used. In these cases, the freedom of the transverse width and height increments of each shell fold forming the transformed sheeting is ensured, and the various attractive and innovative shapes of shell roofs and contraction curves of relatively big curvatures can be achieved (Figure 6) [1,9]. If the fold does not have the freedom of the transverse width increments due to the strong stiffening of its longitudinal edges shared with its adjacent folds, or if the assembly technique causes additional forces varying the effective widths of the fold ends and their supporting lines, the aforementioned interdependence cannot be used.

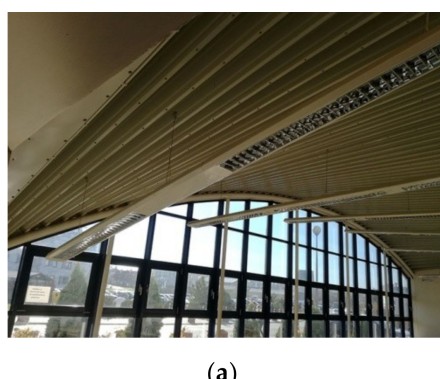
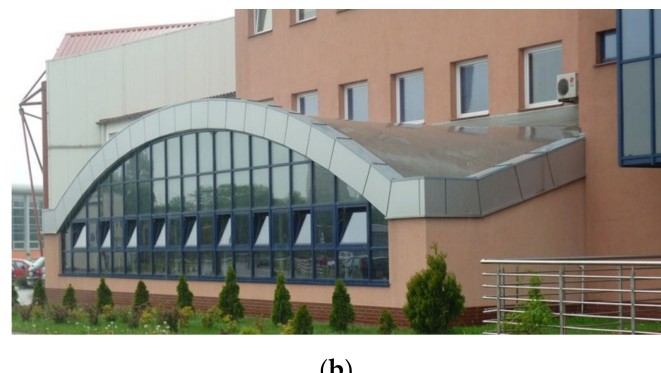

| (**a**) | (**b**) |

**Figure 6.** A transformed roof shell whose inside bottom layer is folded sheeting: (**a**) view from the inside; (**b**) view from the outside.

The application of the well-known conventional design methods [12,13], which is known from the traditional courses of theory of structures, in the shaping of transformed shell roof forms is rather ineffective, because it usually results in high values of normal and shear stresses, local buckling, and the distortion of thin-walled flanges and webs of shell folds. It is often impossible to assemble the designed shell sheeting into skew roof directrices because of the plasticity of the fold's edges between flanges and webs. Reichhart developed various methods for calculating this arrangement and the length of the supporting lines of all the folds in the transformed sheeting [1]. Abramczyk improved the method [8,9] so that the transformation would cause the smallest possible initial stresses of the shell folds.

Therefore, the designer may have to face, and cope with, some problems that arise from using unconventional methods for shaping the general architectural forms of buildings roofed with transformed folded steel sheets, and striving for the relatively simple implementation of the designed innovative forms. The solution of these problems is the priority. The main task is to achieve the geometrical, architectural, and structural cohesion of all the elements of each free-form building, and its shell roof in particular [14]. This aim is accomplished by creating a parametric description of such building free forms and, in the near future, their specific structural systems [15,16] based on the geometrical and mechanical characteristics of the transformed sheeting [10].

Other difficult issues related to the shape transformation of thin-walled folded shells are the diversified supporting conditions, which are calculated for subsequent folds in the same shell. The diversification results from the mutual skew position of the roof directrices, which results in different twist degrees for the subsequent folds in the shell. Thus, the twist degree is the basic border condition that is calculated for each fold in a roof shell, and affects the shell shape of the fold. The aforementioned interdependence between the supporting conditions and the shape of each shell fold is reduced to the interdependence between its twist angle degree and supporting line length. The lengths of the supporting lines of all the subsequent folds along each directrix have basic significance when searching for the fold's shell shapes.

An additional complication is caused by none of the directrices in relation to the transformed shell sheeting needing to be symmetrical nor congruent. This means that the sum of the calculated lengths of the supporting lines of all the subsequent shell folds may be different from the length of both employed directrices, so one of the directrices does not have to be completely covered with the sheeting. The differences can reach more than one meter. In this case, changing at least one of the parameters used in the presented description allows the shape and length of one of the directrices to be adjusted to the width of the whole roof shell. Attempts to change the widths of the fold's transverse ends during their assembly in the shell roof without recalculations are unjustified, because they cause an unnecessary increase in the initial stresses and most often need high forces that can even result in the plasticity of the fold's flanges and webs. This increase may also be a result of miscalculations related to the optimal fold's transformed shapes.

For engineering developments, each shell fold can be modeled with a simplified smooth sector of a warped surface [17,18]. The sum of all such sectors is a continuous edge structure. On the basis of this structure, one single smooth shell sector approximating this structure and modeling the entire transformed roof shell is created. In this case, the loft function of many graphics computer programs can be used.

## 2. Critical Analysis

The two straight lines shown in Figure 7, *x* and *y*, are two rulings of a specific kind of hyperbolic paraboloids. This type of hyperbolic paraboloids, which is characterized by such lines being perpendicular to each other, is often used to model the transformed corrugated shells, which are called hypars. Figure 7a shows a simplified, smooth model of a transformed corrugated shell. The shell has quite unique general geometrical properties that are similar to the central sector of the hyperbolic paraboloid symmetric about the *x* and *y* axes of the Euclidean coordinate system [*x, y, z*]. These axes belong to two various families of rulings of the paraboloid, and divide this paraboloid into four congruent units, which are designated as one, two, three, and four. The dimensions of this central sector are represented by the capital letters A, B, and C. These dimensions are the absolute values of the coordinates of four vertices belonging to the edge line of the central section in [*x, y, z*].

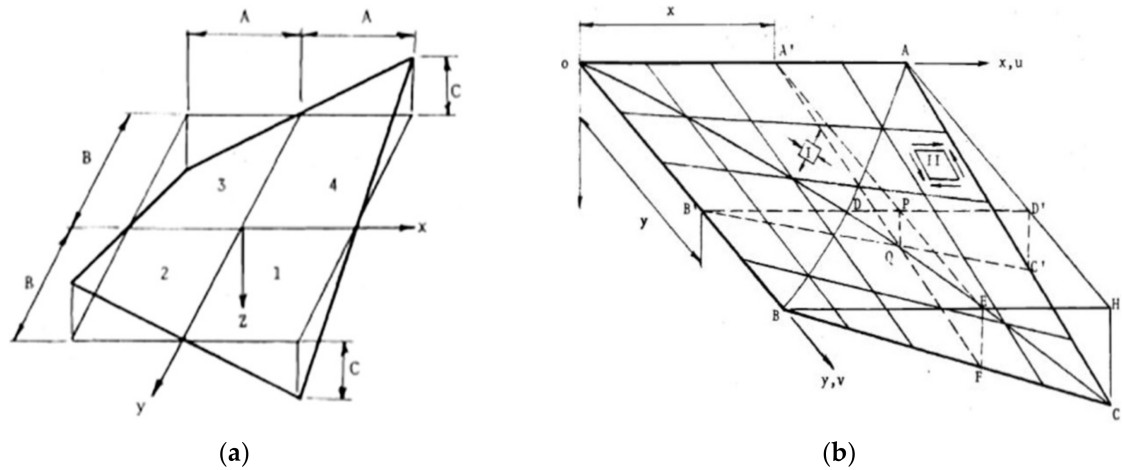

(**a**)　　　　　　　　　　　　　　　　　　　　　(**b**)

**Figure 7.** Basic hypar units: (**a**) a central sector of hyperbolic paraboloid; (**b**) a complete unit: one-fourth of the central sector.

Such surfaces were employed by, for example, McDermott et al. [13,19]. Corrugated shell roofs can be shaped as central sectors, or one-fourth of the central sectors of hyperbolic paraboloids (Figures 7 and 8). Various configurations of shell structures composed of such sectors were used by Fisher et al. [20,21]. The diversity of shell roof forms may be slightly improved by using the computer program developed by Gioncu and Petcu [22].

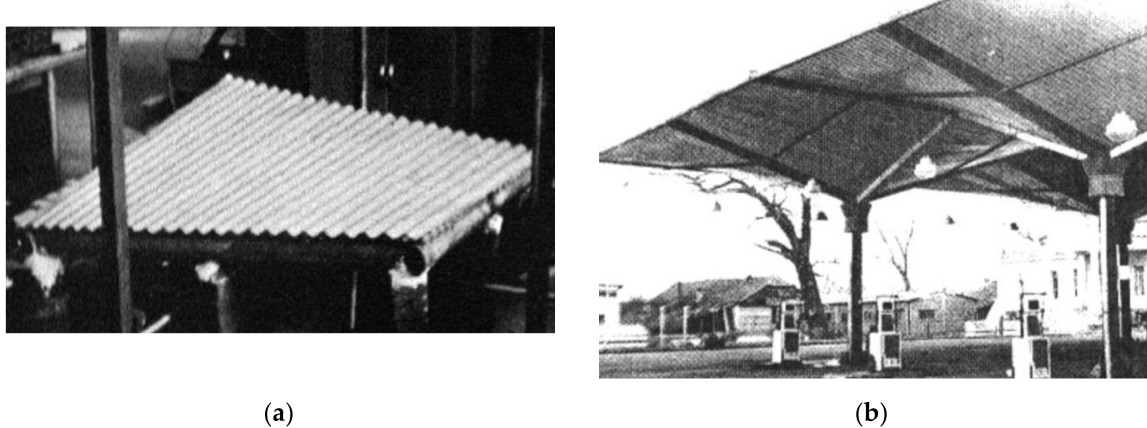

(**a**)　　　　　　　　　　　　　　　　　　　　　　　　　　(**b**)

**Figure 8.** (**a**) The experimental hyperbolic paraboloid shell; (**b**) the erected hyperbolic paraboloid shell structure.

The open thin-walled profile and orthotropic properties of the transformed sheeting result in many advantages and disadvantages of the discussed architectural free forms. This makes it necessary for the shell shape of each fold in the shell sheeting to be optimized in relation to the supporting conditions to obtain the lowest possible negative stresses and strains, as well as attractive shell shapes [8,12].

Based on his experimental tests [1], Reichhart developed an innovative simple method for shaping free-form roofs made up of elastically transformed folded sheets. The concept of his method lies in using the geometrical and mechanical properties of nominally plane-folded steel sheets transformed into rational shell roofs [23]. He employed some basic characteristics of such transformed sheeting observed during his experimental tests. Reichhart's experimental sheets were supported by straight directrices [24]. Kiełbasa created a computational folded model of freely twisted sheets using Reichhart's concept [25].

Abramczyk found Reichhart's concept a very rational approach [8]. However, he has proved that the simplifications made by Reichhart cause very significant errors in roof shell shaping, because they lead to ineffective forced shape transformations and induce unnecessarily high stresses and even plastic deformations of the shell fold's walls.

Computer programming enables the search for innovative diversified corrugated shell roofs and entire building forms [16]. Taking advantage of this possibility, Abramczyk developed a method for the intuitive shaping of free-form buildings covered with plane glass elevations and transformed shell steel roofs [14], and the creation of their simplified models. His method is constantly evolving [26,27], used by graduates [28,29], and extended to complex free-form structures [30] (Figure 9). Some of Abramczyk's tests and analyses were carried out on his computational folded models [10] created in a numerical program called Advances in Dynamic Incremental Nonlinear Analyses [11].

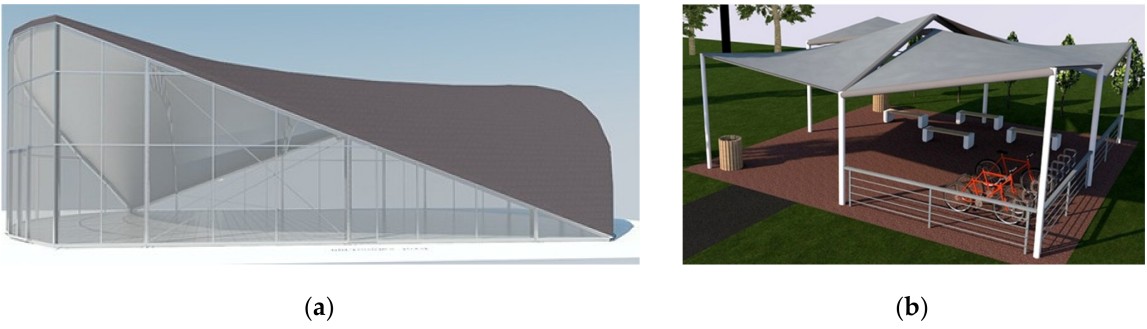

(**a**)　　　　　　　　　　　　　　　　　　　　　　　　　　(**b**)

**Figure 9.** Architectural phase of: (**a**) a complete free-form building, (**b**) a shell roof structure composed of four hyperbolic paraboloids.

### 3. Aim and Scope

The aim of the paper is to carry out and present a novel parametric description of the process of shaping the following:

1. Shell roof forms made up of nominally flat thin-walled folded sheets transformed into spatial forms as a result of connecting their longitudinal edges to obtain a single corrugated strip and arranging the strips on two directrices,
2. General architectural forms of whole buildings using the above transformed sheeting to obtain attractive shapes, dimensions, proportions, and slopes of all the complete building elements such as façades, eaves, and roofs, and their characteristic lines, planes, and surfaces. The goal of this process is to achieve an internal shape integration of the shaped architectural form.

The proposed parametric description allows two things. Namely, complex solutions of the problems presented in the previous sections can be simplified, and, second, a technically useful algorithm that is implemented in computer programs and developed by one of the authors can be created. A method based on the above description and supported by the aforementioned computer programs assists the designer in the process of searching for the expected diversified architectural free forms.

In the article, the authors assume that the proposed parametric description must be presented for an example regarding the search for a visually appealing and internally consistent architectural form characterized by a relatively free general form roofed with a transformed roof sensitive to harmonious incorporation into the expected built environment. For this purpose, it was decided that the object of the search is the simplest set of parameters chosen from all the parameters employed in the presented parametric description. Moreover, the values can be assigned to the selected parameters so that the internal integrity and external sensitivity of the searched form is achieved. Additionally, it was assumed that:

1. The directrices of the roof shell are curved,
2. The curvatures of the transformed shell have to be big enough to achieve the appropriate dimensions of its parts, which should be visible from the directions parallel to the horizontal building base plane and constitute a significant part of the architectural form that is sought.

### 4. Concept

In order to achieve the objectives proposed in the previous section, the following concept of activities is adopted. In the first step, a parametric description of the considered general building forms is used. This step results in a simplified, flat-walled model $\Sigma$ consisting of four quadrangles that have vertices at $P_i$, $B_i$ ($i$ = 1 to 4) and represent the four elevation walls of a building (Figure 10). Both of the $\Sigma$ forms shown in Figure 10 have to be built on the basis of the earlier created reference tetrahedrons $\Gamma$, which are defined by the means of four adopted vertices, $H_i$. One edge of each of these quadrangles is also a segment of a spatial closed line $B_1B_2B_3B_4$, which is a model of straight roof eaves. When the roof directrices $e$ and $f$ are curved, they are usually adopted in planes $\gamma_1$ and $\gamma_3$ (or $\gamma_2$ and $\gamma_4$) of the opposite elevation walls, as shown Figure 10b. It is often assumed that two sides of line $B_1B_2B_3B_4$ are the chords of the adopted arcs $e$ and $f$.

In the second step of the algorithm, a parametric description of the smooth shell models of the building's roofs is used. The accuracy of the models is satisfactory for engineering developments.

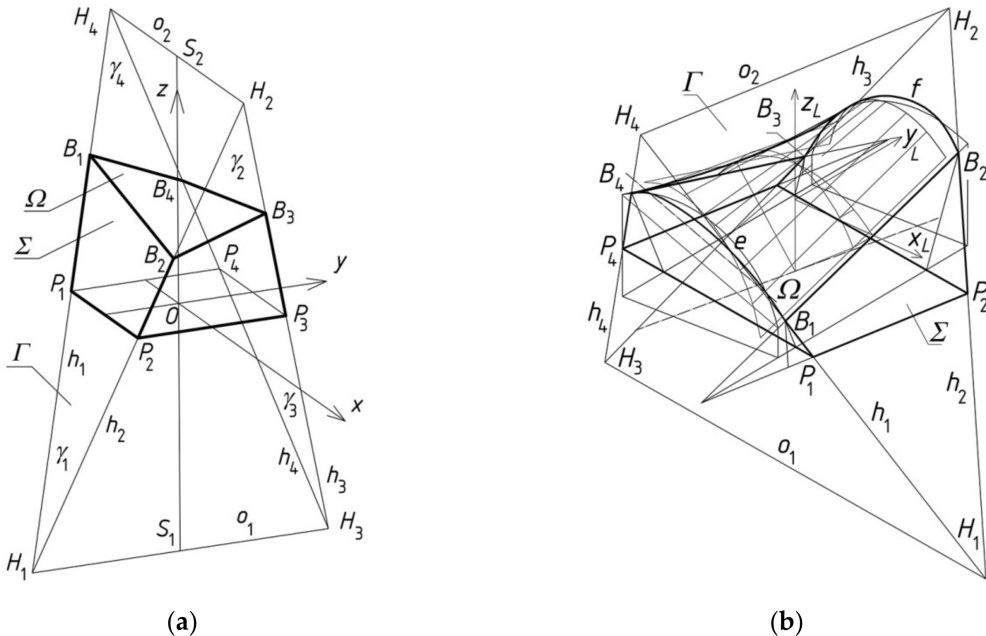

**Figure 10.** Free-form shaping using: (**a**) straight directrices; (**b**) curved directrices *e* and *f*.

In the third step of the algorithm, a parametric description of the basic elements of the building such as flat-walled elevations and shell roofs is used. It includes the thickness of elevation walls and roof shells, the division of each elevation wall into areas creating regular patterns, and the protrusion of roof eaves outside the outline of the elevations.

In the final step of the algorithm, a parametric description of the building structural system dedicated to the folded shell roof and oblique flat-walled glass elevations is used. The description of this step goes beyond the scope of the present paper. It is also possible to extend the method to structures composed of several individual free forms that share walls, such as the ones presented in Figure 11.

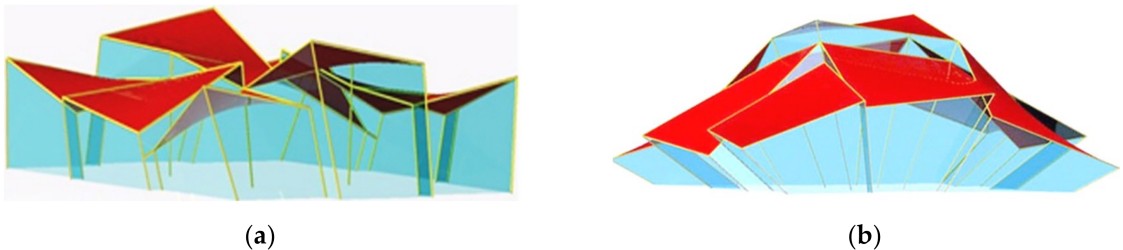

**Figure 11.** Various types of free-form structures. (**a**) Configuration 1; (**b**) Configuration 2.

## 5. Parametric General Building Free Forms

Four flat quadrangles Σ and one sector Ω of a warped surface, which are shown in Figure 12, are the basic objects that are built in this step of the algorithm. They create a simplified model representing the general form of a free-form building. When its roof directrices are two curves, they are the lines limiting two of the aforementioned quadrangles modeling two opposite elevation walls, as shown in Figure 12b. These forms belong to the second basic kind of the architectural free forms discussed in the paper.

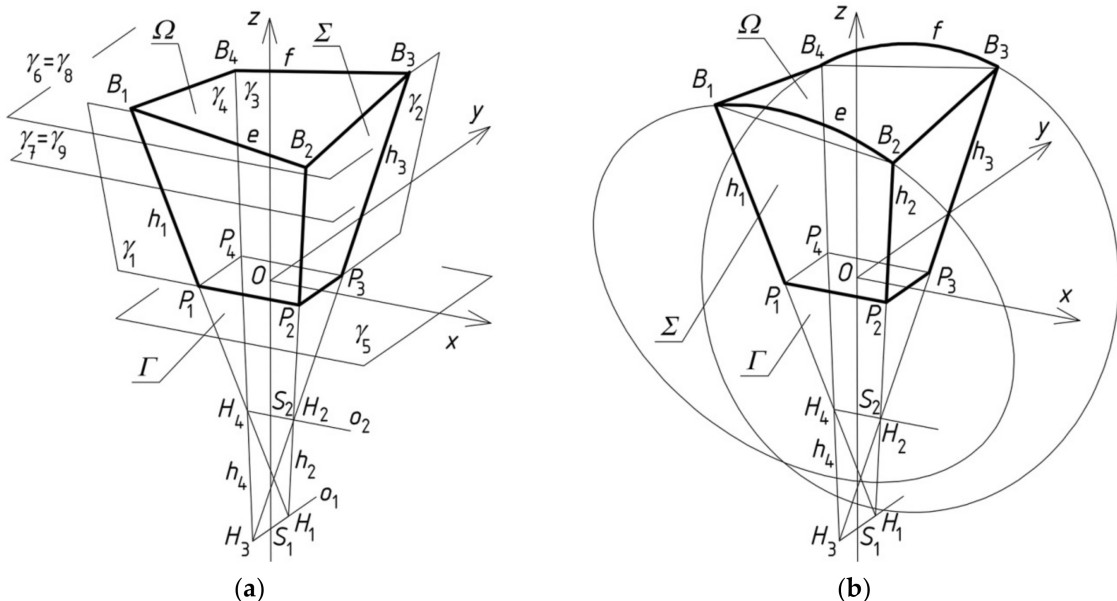

**Figure 12.** Two simplified models of a free-form building roofed with transformed shells supported by: (**a**) straight directrices; and (**b**) curved directrices.

Nine planes $\gamma_i$ ($i$ = one to nine) are the auxiliary objects in modeling the general building form. The first four planes (for $i$ = one to four) allow the construction of the aforementioned quadrangles $P_iP_{i-1}B_{i-1}B_i$ as follows. Each two adjacent quadrangles have a common edge contained in the straight line $h_i$ ($i$ = one to four), which is the intersection of two adjacent planes $\gamma_{i-1}$ and $\gamma_i$. For example, the $P_1P_2B_2B_1$ and $P_2P_3B_3B_2$ quadrangles, which are shown in Figure 12, have the common edge $P_2B_2$ contained in $h_2$. These four planes $\gamma_i$ ($i$ = one to four) define the so-called reference tetrahedron $\Gamma$. The plane $\gamma_5$ contains the building's horizontal base $P_1P_2P_3P_4$. The planes $\gamma_j$ ($j$ = six to nine) define the levels of the corners $B_i$ ($i$ = one to four) of the building eaves. These points belong to $h_i$, too. The opposite planes $\gamma_i$ and $\gamma_{i+2}$ intersect in the axes $o_1$ or $o_2$ of $\Gamma$. The neighboring edges $h_i$ intersect at the vertices $H_i$ of the reference figure $\Gamma$.

The activities carried out on the aforementioned facilities include:

- adopting an orthogonal coordinate system [$x$, $y$, $z$] in three-dimensional space,
- accepting any two points $S_1$ and $S_2$ on the axis $z$,
- passing axis $o_1 \parallel y$ through point $S_1$,
- passing axis $o_2 \parallel x$ through point $S_2$,
- selecting vertices $H_i$ ($i$ = one to four) on axes $o_1$ and $o_2$,
- defining each straight line $h_i$ by means of vertices $H_i$,
- obtaining planes $\gamma_i$ ($i$ = one to four) defined by the respective pairs of neighboring $h_i$,
- creating corners $P_i$ of the rectangular building base as the points of the intersection of plane $\gamma_5$ with each $h_i$,
- determining all the corners of the roof eaves, $B_i$, as the points of the intersection of planes $\gamma_j$ ($j$ = six to nine) with $h_i$, and
- defining directrices $e$ and $f$ contained in any two opposite planes $\gamma_1$ and $\gamma_3$ or $\gamma_2$ and $\gamma_4$.

The following parameters describing the building general forms were adopted in the algorithm:

1. $ptr_k$ (for $k$ = one to six) representing the lengths of the sections: $S_2O$, $S_1O$, $S_1H_1$, $S_1H_3$, $S_2H_2$, and $S_2H_4$,
2. $ptr_r$ (for $r$ = seven to 10) representing the distances of planes $\gamma_j$ ($j$ = six to nine) from plane $\gamma_5(x, y)$ of the building base,

3.  $ptr_{10}$ representing the same ridges of roof directrices $e$ and $f$, which are usually shaped in the form of circle arcs.

It is assumed that the architectural form example belonging to the last—that is, third—basic type (Figure 13), which is discussed below, will be determined on the basis of the parametric description proposed above. The values of the parameters adopted for shaping the sought architectural form are given in Table 1. From the engineering point of view, what is more important is the coherence of the parametric description of architectural forms and the obtained set of proportions between the parameters, rather than the individual values of these parameters. In order to present these proportions, it is assumed that the reference parameter is the width $ptr_{13}$ of the general form along its base. The values of the proportions that are considered important and shown in Table 2 refer to this reference parameter.

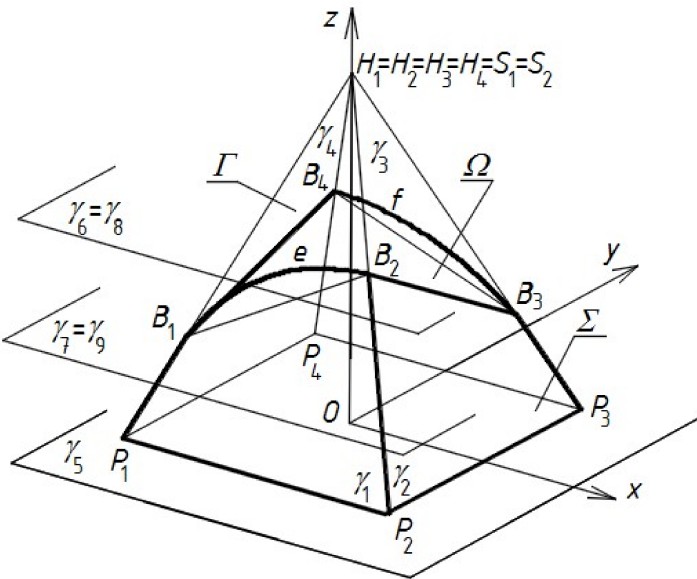

**Figure 13.** Simplified model of a building free form of the third type roofed with a transformed shell.

**Table 1.** Parameters adopted for the examined architectural free form from Figure 13.

| Parameter | Value |
|---|---|
| $ptr_1 = ptr_2$ | 23,333.3 |
| $ptr_i$ for $i = 3$ to 6 | 0.0 |
| $ptr_7 = ptr_9$ | 12,746.2 |
| $ptr_8 = ptr_{10}$ | 12,746.2 |
| $ptr_{11}$ | 1790.9 |
| $ptr_{12}$ | 726.7 |
| $ptr_{13}$ | 20,000.0 |

[1] values in millimeters.

**Table 2.** Proportions calculated for the examined architectural free form from Figure 13.

| Proportion | Value |
|---|---|
| $ptr_1/ptr_{13} = ptr_2/ptr_{13}$ | 1.17 |
| $ptr_i/ptr_{13}$ for $i = 3$ to 6 | 0.0 |
| $ptr_7/ptr_{13} = ptr_9/ptr_{13}$ | 0.64 |
| $ptr_8/ptr_{13} = ptr_{10}/ptr_{13}$ | 0.64 |
| $ptr_{11}/ptr_{13}$ | 0.090 |
| $ptr_{12}/ptr_{13}$ | 0.036 |

In order to explain the attractiveness of a general form Σ, it is necessary to define additional parameters describing, for example, the width and height of the created architectural form, its roof and façade, the slopes of the roof eaves, edges, and the planes of façade walls. Such an action requires extensive considerations, and does not fall within the scope of the article. These issues are initially discussed in [13], and are related to a specific method, which is not presented in this article.

It is possible to create other sets of parameters defining the general building forms, as in the examples proposed by Abramczyk in [8,26]. However, his method is significantly more complex and requires a good spatial reasoning from the designer.

## 6. Parametric Shell Roofs

### 6.1. Introduction

Each regular warped surface [8,31] has straight rulings $t_i$ and a line of striction $s$ that intersects all of the rulings at the so-called central points $S_i$. When two rulings $t_{i-1}$ and $t_i$ of the surface approach each other at an infinitely short distance, then point $S_i$ is the nearest point of $t_i$ relative to ruling $t_{i-1}$. Line $s$ is simply a contraction of the warped surface (Figure 14).

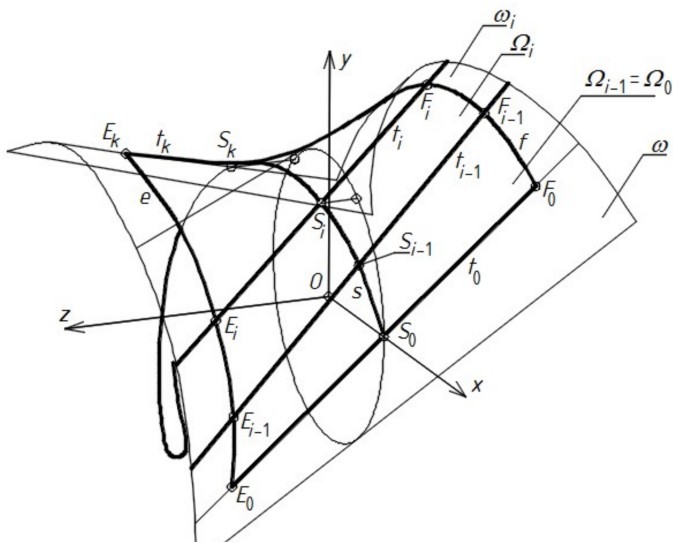

**Figure 14.** Sector Ω of warped surface $\omega$ modeling effectively transformed shell roof by means of contraction helix $s$ ($S_0, \dots, S_i, \dots, S_k$) of Ω.

It assumed that the created simplified models of all the discussed shell folds (possibly the entire single sheets if the curvatures of the considered shells are small) transformed into shell shapes are the central sectors $\Omega_i$ of different warped surfaces. In each such sector, there is a striction line $s$ passing transversally, halfway along its length. Each sector is limited by rulings $t_{i-1}$ and $t_i$ lying in a proper distance from each other; the edge line of $\Omega_i$ is composed of the $E_{i-1}E_i$ section of directrix $e$, the $E_iF_i$ section of ruling $t_i$, the $F_iF_{i-1}$ section of directrix $f$, and the $E_{i-1}F_{i-1}$ section of ruling $t_{i-1}$.

Despite the attempts [8], it is impossible to invent one general mathematical equation defining all of the types of the warped surfaces used in modeling the discussed transformed roof shells. Therefore, all of the rulings $t_i$ have to be determined in an approximate way on the basis of the adopted directrices $e, f$, and ruling $t_{i-1}$, which is calculated either at the previous step or adopted as $t_0$ at the beginning. The procedure of the latter solution is as follows. It is assumed that the $E_{i-1}$, $F_{i-1}$, and $R_{i-1}$ points were constructed in the previous step, or $E_{i-1} = E_0$, $F_{i-1} = F_0$, and $R_{i-1} = R_0$ were calculated in the first step (Figure 15).

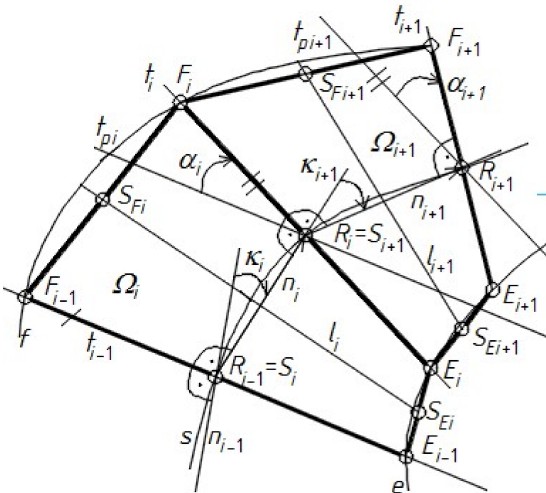

**Figure 15.** Shell sectors $\Omega_i$ and $\Omega_{i+1}$ modeling two subsequent effectively transformed shell roof folds limited by pairs of skew straight lines $\{t_{i-1}, t_i\}$ and $\{t_i, t_{i+1}\}$, which are determined on the basis of the lines $n_i$ normal to the rulings $t_i$ and $t_{i-1}$.

On the basis of these points, the positions of points $E_i$ and $F_i$ on directrices $e$ and $f$ and ruling $t_i$ ($E_i$, $F_i$) are determined so that the straight line $n_i$ perpendicular to $t_{i-1}$ and $t_i$ intersects line $t_{i-1}$ at point $R_{i-1}$, and the area of the sector $\Omega_i$ modeling a shell fold after transformation is equal to the surface area of the same fold before transformation. The last condition can be used with satisfactory accuracy for engineering developments, but not for scientific research [8,10].

The positions of points $E_i$ and $F_i$ and ruling $t_i$ are determined by the surface area and the twist angle of each transformed fold. The twist angle $\alpha$ of fold $\Omega_i$ is the angle of inclination of two planes: $(S_{Ei}, S_{Fi}, E_i)$ and $(S_{Ei}, S_{Fi}, F_i)$. The unit twist angle $\alpha_j$ of the fold is expressed as the quotient of the above twist angle $\alpha$ by length $|S_{Ei}S_{Fi}|$ of the fold. The degree of the fold's twist is the measure of the unit twist angle $\alpha_j$, which is regarded as constant at the length of the fold. The length of the fold is taken as the length of the $S_{Ei}S_{Fi}$ section, where $S_{Ei}$ and $S_{Fi}$ are the midpoints of the $E_{i-1}E_i$ and $F_{i-1}F_i$ sections. Particular attention should be paid to the variation in the length and twist angle of the subsequent folds in a shell. The unit twist angle represents the basic geometrical supporting condition of the fold. Variable fold lengths indicate that the transverse ends of these folds have to be cut differently to adapt these ends to the directrices' directions.

Figure 15 shows the method of the shell fold's modeling by means of a special type of warped surfaces, i.e., such surfaces whose rulings are perpendicular to line $s$ of striction. In a general case of a warped surface, its rulings are not perpendicular to its line of striction. Consequently, the subsequent straight lines $n_i$ and $n_{i+1}$ do not intersect the ruling $t_i$ at the same point, which results in displacing points $S_{i+1}$ and $R_i$ along ruling $t_i$ (Figure 16).

In this case, the procedure of searching for a simplified smooth model $\Omega_i$ of an effectively transformed shell fold should be extended by a second condition, in addition to the condition concerning the equality of surface areas of the fold before and after its transformation. This condition concerns the minimization of the length of the $R_iS_{i+1}$ segment and positioning points $S_{i+1}$, $R_i$ as close as possible to the midpoint of $E_iF_i$ by looking for appropriate proportions between the lengths of the $E_{i-1}E_i$ and $F_{i-1}F_i$ lines. Consequently, the parametric description of shaping all of the shell folds has to cover both of the basic conditions that determine the efficiency of transformations of all the modeled shell folds in the transformed roof.

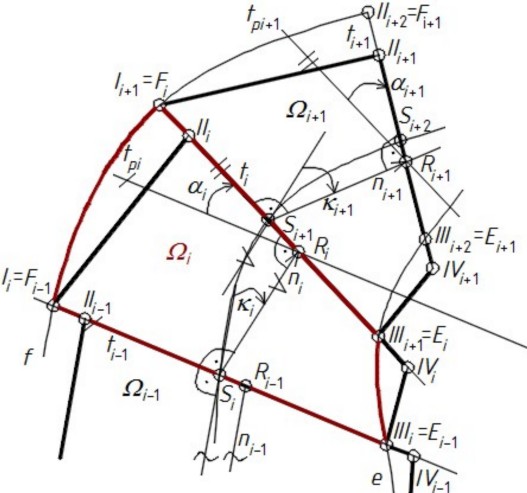

**Figure 16.** Models of two effectively twisted, bent, and sheared subsequent shell folds limited by pairs of two skew straight lines $\{t_i, t_{i+1}\}$, which are determined on the basis of a discontinuous sum of many $R_iS_i$ straight sections.

## 6.2. Simplified Smooth Parametric Models for Corrugated Roof Shells

The two conditions presented in the above introduction are implemented in the Rhino/Grasshopper program, which is orientated to the parametric modeling of geometric objects (Figure 17) and written by one of the authors. All of the individual objects and operations on these objects that are performed following the algorithm are defined by means of the flat rectangular graphic elements, which are named components, and are positioned on the named canvas of the Grasshopper background. The relationships between these objects are described by means of lines called connectors or wires. The application assists with creating inseparable simplified smooth shell models of all the subsequent shell folds (Figure 18). On the basis of the edge sum, a single smooth surface modeling of an entire building shell roof can be built.

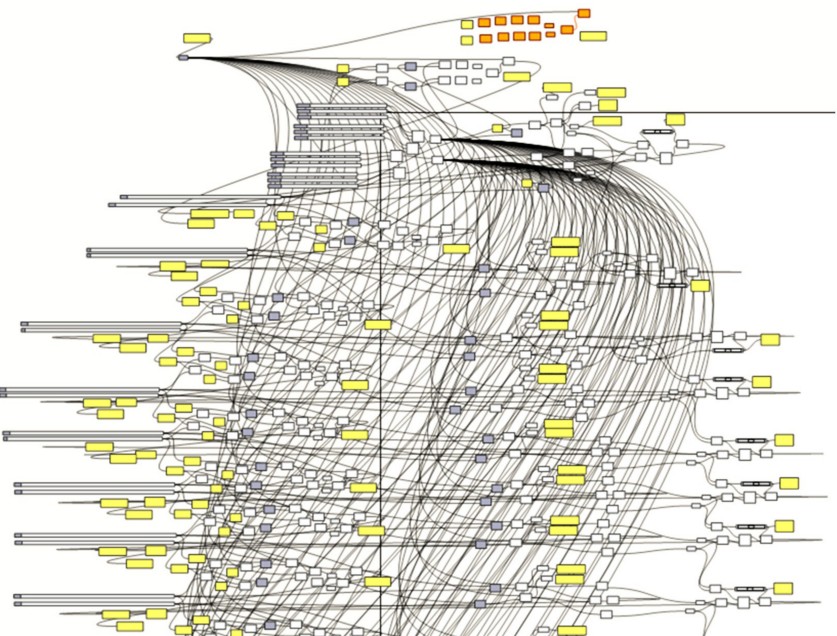

**Figure 17.** Part of the scheme of many objects creating the parametric algorithm implementing the authors' parametric description.

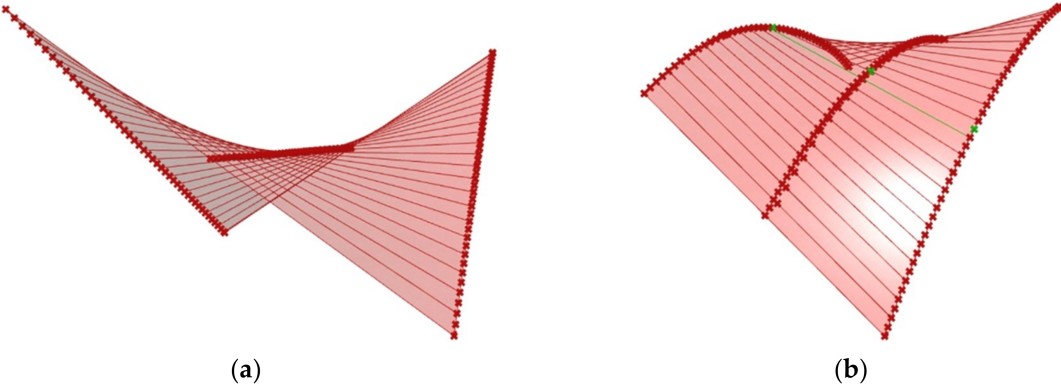

**Figure 18.** Simplified, smooth models of two transformed shells built in the Rhino/Grasshopper program and defined with: (**a**) straight directrices; and (**b**) curved directrices.

This step of the method's algorithm results in a smooth shell model of transformed roof sheeting. It is a sector of a warped surface, and can be formed as one of the two shapes presented in Figure 18. The geometrical properties of these smooth sectors represent the complex spatial deformed forms of all the folds of the transformed sheeting in a simplified way [23]. The current step ensures that the geometrical characteristics of these elements take account of the geometrical and structural properties of thin-walled folded sheets transformed in experimental tests, and their accurate models used in computer simulations carried out in the ADINA program [10].

The authors' application for the Rhino/Grasshopper program makes it possible to define two directrices $e$ and $f$ as algebraic lines, using two tetrads of the adopted points belonging to these directrices, whose coordinates are the entered initial data. If these directrices are straight, the coordinates of both their ends may only be entered. In this case, the four triads of sliders that are needed (Figure 19) to enter the coordinates of the ends of $e$ and $f$ are determined. Two components generating these straight directrices are also shown in Figure 19.

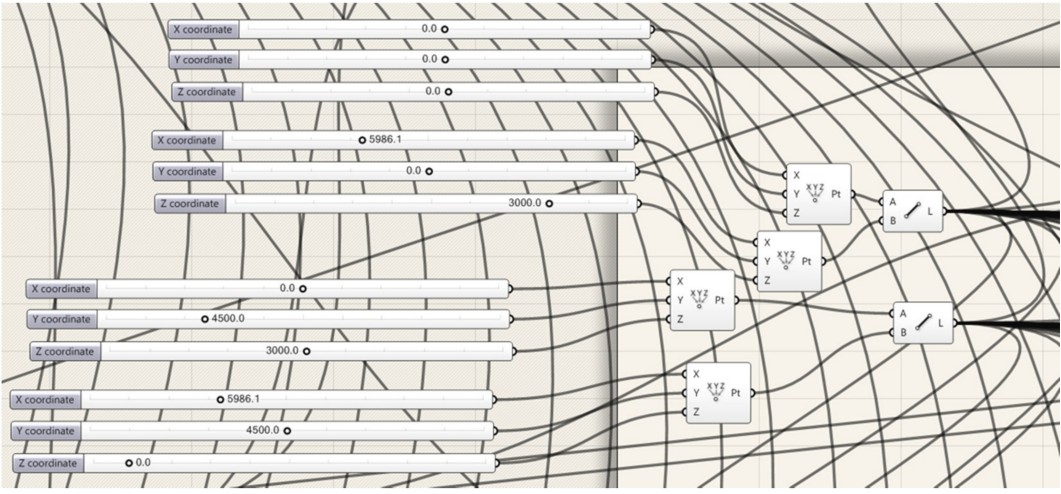

**Figure 19.** Four triad of sliders that allow three coordinates of the ends of two directrices, $e$ and $f$, to be entered.

The first three sliders shown at the top of Figure 19 define three coordinates of the starting point of the first directrix $f$. The second three sliders allow entering the coordinate of the end point of $f$. The two next triads of sliders define the coordinates of the starting and end points of the second directrix, $e$.

Segments $e_i$ and $f_i$ of directrices $e$ and $f$ are the auxiliary objects of this step (Figure 20). They correspond to the supporting lines of the subsequent folds of a shell roof. The lengths of these segments

are calculated, while the simplified shell model of each roof shell fold is developed. A central sector $\Omega_i$ of a warped surface limited by two rulings $t_{i-1}$ and $t_i$ and modeling this shell fold is sought by the means of points $E_i$ and $F_i$ displaced on $e$ and $f$ by means of the next two sliders, which are presented in Figure 21.

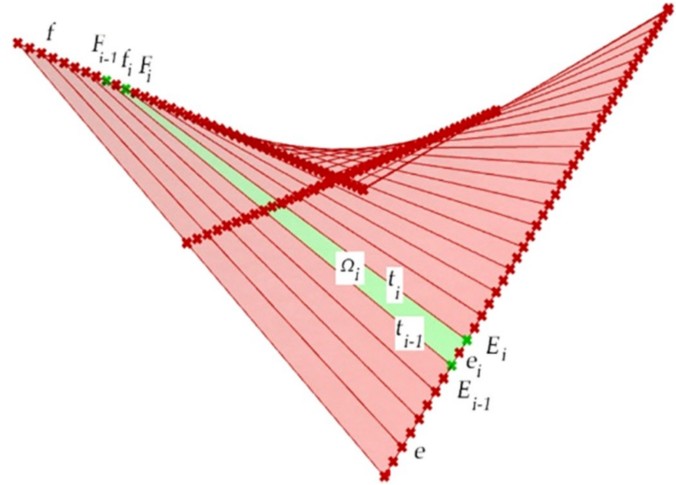

**Figure 20.** Narrow smooth shell sector $\Omega_i$ modeling a complete shell fold, which is created as the loft component and limited by two rulings, $t_{i-1}(E_{i-1}, F_{i-1})$ and $t_i(E_i, F_i)$, as well as two curves, $e_i$ and $f_i$.

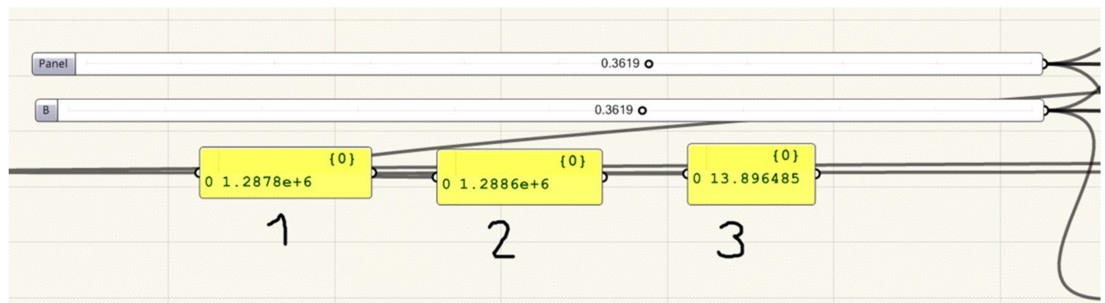

**Figure 21.** Two sliders and three panels assisting in meeting two conditions for effective fold transformations.

In the left panel signed '1', the information about the surface area (in millimetres) of the simplified model of the investigated fold before transformation is provided. In the middle panel signed '2', information on the surface area of the simplified model of this fold after transformation is given. The right-hand panel named '3' provides information about the distance between the contraction lines of the simplified models of two adjacent folds after transformation, which is the length of segment $R_i$ and $S_{i+1}$, which was discussed in the introduction (Figure 16). The optimization of the shell shape of each designed fold in the shell consists of a change in the values of the above sliders such that the two numbers shown in panels '1' and '2' are equal to each other with the accuracy of about $10^4$ mm, while the number in panel '3' should be close to zero, with the accuracy of up to 50 mm.

The two sliders presented in Figure 21 control the position of $E_i$ on $e$ and $F_i$ on $f$, so they decide of the shape of $\Omega_i$. The change of the values of these sliders, which causes a change of the values in panels '1' to '3', results in satisfying the two main conditions listed in the introduction, and is related to the surface areas of the simplified models of transformed folds and the optimal position of the contraction lines along the length of the folds. The conditions determine whether the modeled fold is effectively transformed or not.

Following the algorithm developed by the authors, the simplified model of each shell fold has to meet two conditions determining the harmonious and optimal work of all the folds in a

shell. As we know, one concerns the strictly defined surface area of the smooth shell model of each shell fold in relation to its geometrical supporting conditions. It is represented in the authors' program by the green container shown on the left in Figure 22. The other condition concerns the contraction of each transformed corrugated shell. This condition requires that the contraction line passes transversely in relation to the directions of the shell folds through the middles of these folds along their lengths. The contraction is designated as *s* in Figures 14 and 15. In the authors' application of the Rhino/Grasshopper program, this condition is represented by the green container shown on the right in Figure 22. This container makes it possible to find rulings $t_{i-1}$ and $t_i$ perpendicular to contraction line *s*, and the distance between the $R_i$ and $S_i$ points that were discussed in the introduction.

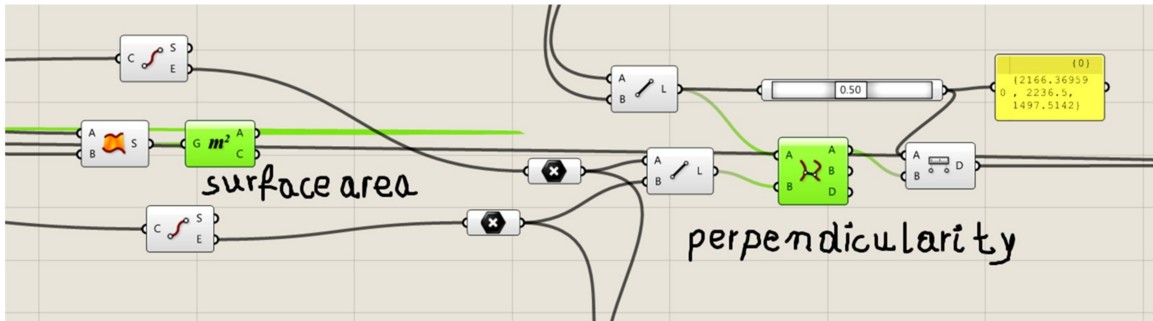

**Figure 22.** Two basic green components representing two basic conditions related to the fold's surface areas and line of contraction.

The two lines $E_0E_i$ and $F_0F_i$ shown in Figure 14 and called subcurves, and are used in the construction of $E_i$ and $F_i$ points on *e* and *f*. They are generated with the components shown in green in Figure 23. The application determines the $e_i$ and $f_i$ edge lines of $\Omega_i$ as differences between the subcurves $E_0E_i$ and $E_0E_{i-1}$, and $F_0F_i$ and $F_0F_{i-1}$, which were calculated for the adjacent folds in the shaped shell.

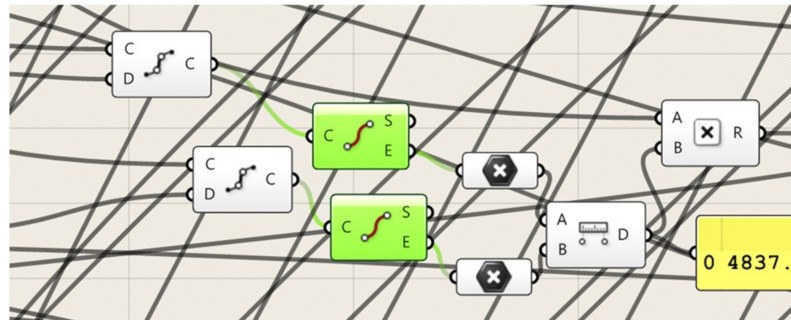

**Figure 23.** Components representing subcurves that are helpful in determining models $e_i$ and $f_i$ of the fold's supporting lines.

As a result of the comparison of the fold's surface areas before and after the shape transformation, the $b_t$ width of each transformed fold along the appropriate directrix is displayed by means of the monitor component shown in Figure 24 as the number 289.944345, which was measured in millimetres. The nominal width of this fold before transformation was 280 mm. It is also displayed on one of the panels shown in Figure 24. The unit twist angle was equal to 8.2111045.

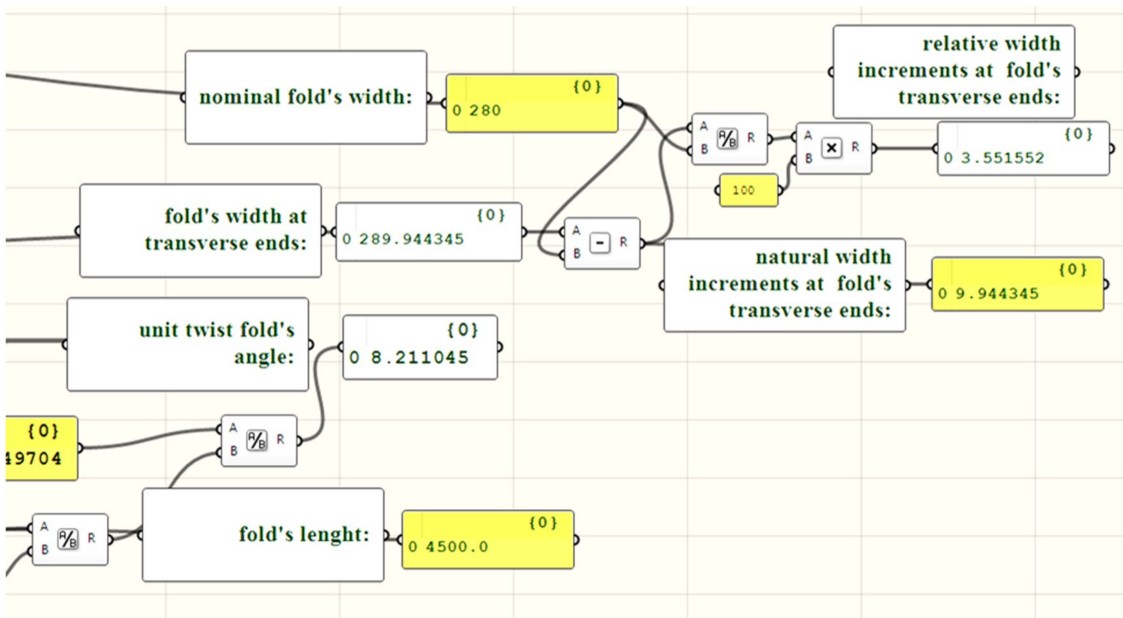

**Figure 24.** Important features of a simplified smooth shell model of the central fold of a shell, whose unit twisting angle $\alpha_j$ is equal to 8.211045°.

For engineering developments, the unit twist angle is the only significant parameter determining the supporting conditions of each transformed fold in a shell. The relative width increments of the shell fold along each directrix are the parameters that decide the shell form of the fold. For the shell fold, whose geometrical properties and supporting conditions are presented in the cells shown in Figure 21, the relative width increments along the both directrices are equal to 3.551552%. The relative increase $b_{wr}$ in the fold's width is the quotient of the absolute width increment $b_w$ of the transformed fold to the width $b_0$ of the fold before the transformation. It can be calculated following the formula:

$$b_{wr} = \frac{b_w}{b_0} \cdot 100\%,\tag{1}$$

In Figure 24, the $b_w$ value is equal to 9.944345, which is the subtraction of $b_0$ = 280 mm from $b_t$ = 289.944345. The $b_w$ absolute width increment of the fold is then the difference between the width $b_t$ of the fold after the transformation, and the $b_0$ value before the transformation. The width $b_t$ of each fold in the shell is calculated as the length of the segment $e_i$ of $e$ or $f_i$ of $f$ (Figure 20).

Most of the relationships that were obtained during experimental tests are non-linear because of the big mutual displacements of adjacent folds, and the considerable deformations of the flanges and webs of these folds in the same shell sheeting. Such a non-linear relationship between the relative width increments the $b_{wr}$ values of various experimental shell folds at their transverse ends, and the measure of the fold's unit twist angle $\alpha_j$ is shown in Figure 25. The obtained dependencies are functional dependencies, and can be used to calculate the width of the transformed folds depending on the unit twist angles of these folds. The similar nature of the curves indicates the relatively insignificant interdependence between the change in the width of the effectively transformed folds and their height.

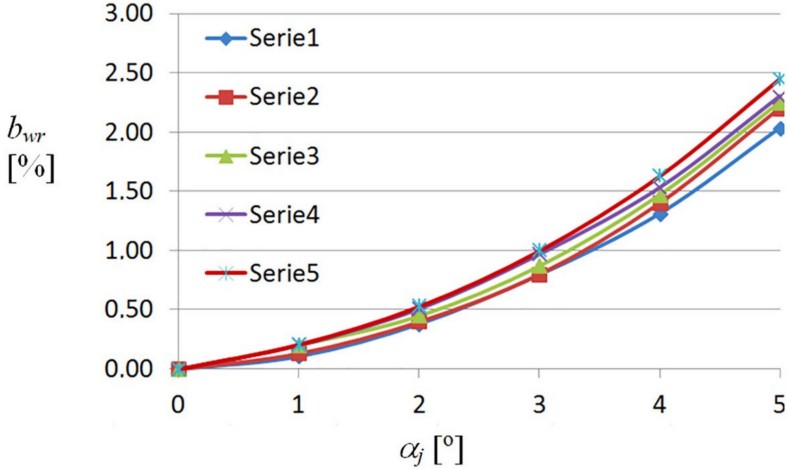

**Figure 25.** Non-linear relationships between the $b_{wr}$ relative width increments of the fold's crosswise ends and $\alpha_j$ unit angle obtained for various heights of the experimental shell folds: Serie1—50 mm; Serie2—55 mm; Serie3—85 mm; Serie4—136 mm; Serie5—160 mm.

In this step, no new parameter, that is no new independent variable, needs to be adopted. The basic activities of this step are:

- determination of the supporting conditions for all the subsequent folds in the shell sheeting, including the fold's twisting angles, based on the mutual position and shape of the roof directrices,
- calculation of the lengths of supporting lines $e_i$ and $f_i$ for each fold on the basis of these supporting conditions,
- calculation of the arrangement of the supporting points of the fold's transverse ends along each directrix and the total length of directrices $e$ and $f$,
- determination of the finite number of rulings corresponding to the longitudinal edges of all the shell folds, and
- a possible correction to the shapes of these directrices to obtain the complete coverage of both roof directrices.

The new dependent variables used in this step of the algorithm include:

- the accuracy of the calculations related to the location of the shell fold contraction at its length, and
- the accuracy of the calculations related to the surface area of the smooth shell model of each investigated fold.

## 7. Parametric Elevation Elements

In the third step of the algorithm, a parametric description of the spatial forms of the considered building roofs and elevations is used. This description also includes the materials from which the roof and elevations are made. Two sufficiently accurate models are shown in Figure 26. The first model, Figure 26a, takes account of the thickness of the roof and elevations, as well as upper, lower, and lateral surfaces of the roof. The other one presents a regular pattern on its elevation walls. The main object of this step is a model taking account of the above-mentioned properties of roof and elevations.

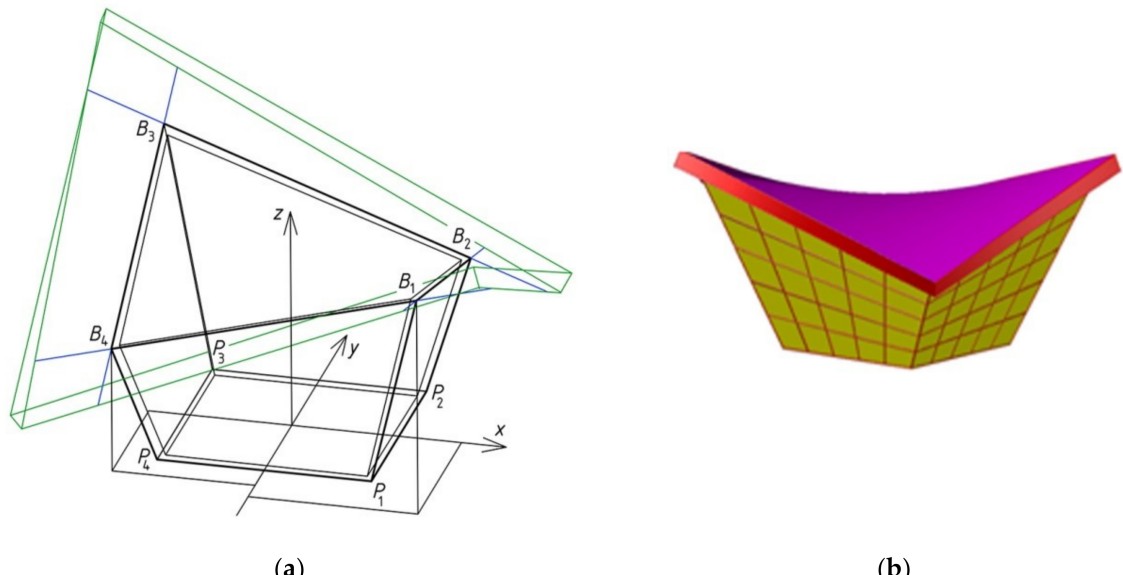

(**a**)                                                                                              (**b**)

**Figure 26.** Two models presenting: (**a**) the thickness of roof and elevations; and (**b**) the regular pattern on elevations.

The main geometrical elements such as surfaces, planes, lines, and points are the auxiliary objects of this step. The intersections, displacements, and rotations of these elements allow all of the building elements as finite sectors, sides, and edges to be modeled and arranged relative to the building construction axes.

The mutual location of the roof and elevations result from the structure and overall dimensions of the general building form. Therefore, the new parameters used in this step are only:

- $ptr_{14}$—the thickness of the roof,
- $ptr_{15}$—the roof overhang outside the outline of elevation walls, and
- $ptr_{16}$, $ptr_{17}$—the pitch, position, and inclination of the regular elevation pattern.

## 8. Example of Shaping Architectural Free Forms

This section presents the geometrical properties of the transformed roof shell roofing for the architectural form that is sought after in this work, the general form of which has been defined using the parameters given in Table 1 in Section 5, and is illustrated in Figure 13.

The shell roof is characterized by non-zero thickness, which is expressed by means of parameter $ptr_{14}$ = 720 mm, and the overhang of the eaves outside the outline of the façade, which is expressed by the parameter $ptr_{15}$ = 500 mm (Figure 27). The coordinates of its characteristic points and the general form of the entire discussed architectural form are given in Table 3. Points $D_{e1}$, $D_{e2}$, $D_{f1}$, and $D_{f2}$ are additionally selected on the $e_g$ and $f_g$ directrices of the upper roof shell surface $\Omega_g$, and their coordinates are entered with the coordinates of points $D_{g1}$, $D_{g2}$, $D_{g3}$, and $D_{g4}$ as input defining the directrices used in the application of the Rhino/Grasshopper program, which was discussed in Section 6. These values were adopted as the suggested values of the sliders that are presented in Figure 19.

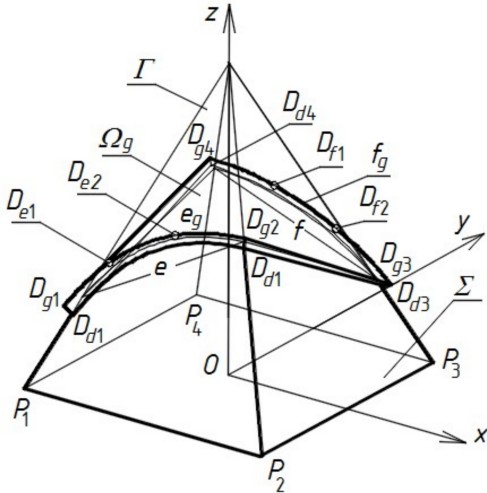

**Figure 27.** Selected points and lines characterizing the general form of the discussed free form.

**Table 3.** Parameters achieved for the examined architectural free form from Figure 27.

| Vertex | X-Coordinate | Y-Coordinate | Z-Coordinate |
|---|---|---|---|
| $P_1$ | −10,000.0 | −10,000.0 | 0.0 |
| $P_2$ | 10,000.0 | −10000.0 | 0.0 |
| $P_3$ | 10,000.0 | 10000.0 | 0.0 |
| $P_4$ | −10,000.0 | 10000.0 | 0.0 |
| $B_1$ | −7190.2 | 7190.2 | 6556.2 |
| $B_2$ | 4537.3 | −4537.3 | 12,746.2 |
| $B_3$ | 7190.2 | −7190.2 | 6556.2 |
| $B_4$ | −4537.3 | 4537.3 | 12,746.2 |
| $S_1 = S_2 = H_1 = H_2 = H_3 = H_4$ | 0.0 | 0.0 | 23,333.3 |
| $D_{g1}$ | −8011.4 | −7977.4 | 5988.8 |
| $D_{g2}$ | 5009.9 | −4885.4 | 13,203.5 |
| $D_{g3}$ | 8011.4 | 7977.4 | 5988.8 |
| $D_{g4}$ | −5009.9 | 4885.4 | 13,203.5 |
| $D_{d1}$ | −7128.7 | −8162.4 | 5556.9 |
| $D_{d2}$ | 5140.9 | −5166.9 | 12,546.4 |
| $D_{d3}$ | 7128.7 | 8162.4 | 5556.9 |
| $D_{d4}$ | −5140.9 | 5166.9 | 12,546.4 |
| $D_{e1}$ | −5349.2 | −6561.3 | 9293.0 |
| $D_{e2}$ | −413.5 | −5291.2 | 12,256.5 |
| $D_{f1}$ | 413.5 | 5291.2 | 12,256.5 |
| $D_{f2}$ | 5349.2 | 6561.3 | 9293.0 |

[1] Values in millimeters.

The discussed roof shell is limited from the top and bottom by two oblique surfaces, the upper one of which is the sought-after model of the transformed folded shell sheeting. This model (Figure 28) was determined using the innovative application built by one of the authors in the Rhino/Grasshopper program.

The calculated values of the unit twist angle $\alpha_j$ defining the supporting conditions, and geometrical properties of the subsequent folds in the discussed shell roof, are tabulated in Table 4.

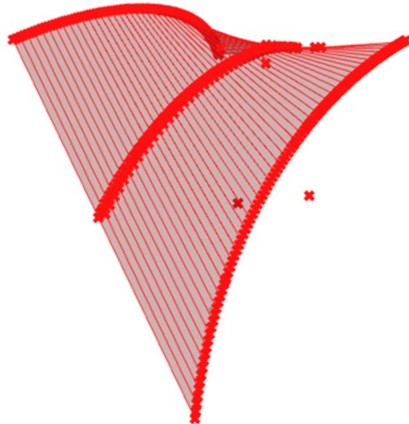

**Figure 28.** Smooth model of the upper surface of the shell roof being sought.

**Table 4.** Parameters describing the subsequent shell folds in the transformed shell whose simplified smooth model is shown in Figure 28.

| Shell Fold [no.] | Length of Supporting Line $e_i$ [mm] | Length of Supporting Line $f_i$ [mm] | Fold's Length [mm] | Fold's Unit Twist Angle $\alpha_j$ [°] |
|---|---|---|---|---|
| 1 | 481 | 314.2 | 14,895 | 3.9834 |
| 2 | 469.5 | 314.3 | 14,595 | 4.037 |
| 3 | 458.6 | 314.5 | 14,313 | 4.0927 |
| 4 | 448,6 | 314.9 | 14,047 | 4.15 |
| 5 | 439 | 315.4 | 13,797 | 4.2081 |
| 6 | 430.1 | 315.9 | 13,563 | 4.2664 |
| 7 | 421.8 | 316.7 | 13,344 | 4.3243 |
| 8 | 414 | 317.4 | 13,140 | 4.3811 |
| 9 | 406.4 | 318.3 | 12,950 | 4.4363 |
| 10 | 399.4 | 319.3 | 12,775 | 4.4893 |
| 11 | 393 | 320.4 | 12,615 | 4.5399 |
| 12 | 386.6 | 321.6 | 12,468 | 4.589 |
| 13 | 381.1 | 322.8 | 12,335 | 4.637 |
| 14 | 375.6 | 324.4 | 12,216 | 4.6835 |
| 15 | 370.7 | 326.1 | 12,110 | 4.7279 |
| 16 | 366.2 | 327.9 | 12,018 | 4.7696 |
| 17 | 361.9 | 329.8 | 11,938 | 4.8082 |
| 18 | 357.7 | 332.2 | 11,872 | 4.8433 |
| 19 | 353.8 | 334.4 | 11,819 | 4.8731 |
| 20 | 350.2 | 336.9 | 11,779 | 4.8963 |
| 21 | 347.1 | 339.5 | 11,752 | 4.9125 |
| 22 | 343.9 | 342.3 | 11,738 | 4.9216 |
| 23 | 340.9 | 345.5 | 11,737 | 4.9234 |
| 24 | 337.9 | 348.7 | 11,750 | 4.9181 |

**Table 4.** *Cont.*

| Shell Fold [no.] | Length of Supporting Line $e_i$ [mm] | Length of Supporting Line $f_i$ [mm] | Fold's Length [mm] | Fold's Unit Twist Angle $\alpha_j$ [°] |
|---|---|---|---|---|
| 25 | 335.4 | 352.2 | 11,775 | 4.9057 |
| 26 | 333 | 355.5 | 11,813 | 4.8862 |
| 27 | 330.6 | 359.5 | 11,865 | 4.86 |
| 28 | 328.6 | 363.8 | 11,930 | 4.8279 |
| 29 | 326.7 | 368.1 | 12,008 | 4.7918 |
| 30 | 324.8 | 372.7 | 12,099 | 4.7525 |
| 31 | 323.2 | 377.8 | 12,204 | 4.7104 |
| 32 | 321.6 | 383.3 | 12,322 | 4.666 |
| 33 | 320.3 | 389.2 | 12,455 | 4.62 |
| 34 | 319.1 | 395.6 | 12,601 | 4.5728 |
| 35 | 318 | 402.4 | 12,761 | 4.5245 |
| 36 | 317 | 409.6 | 12,936 | 4.4737 |
| 37 | 316.1 | 417.2 | 13,125 | 4.4206 |
| 38 | 315.2 | 425.6 | 13,330 | 4.3658 |
| 39 | 314.5 | 434.3 | 13,550 | 4.31 |
| 40 | 314 | 443.5 | 13,786 | 4.2534 |
| 41 | 330.8 | 427.6 | 14,026 | 4.1907 |
| 42 | 312.7 | 462.3 | 14,282 | 4.1288 |
| 43 | 312.5 | 473.6 | 14,566 | 4.0746 |
| 44 | 294.7 | 513.9 | 14,880 | 4.0292 |

The final architectural free form that is sought is presented in Figure 29. The parameter $ptr_{16}$, which is defining the position of the subsequent horizontal lines in the regular elevation pattern, is 3000 mm. The parameter $ptr_{17}$, which is defining the position of the vertical lines of the regular elevation pattern, is four, because of four vertical elevation glass strips.

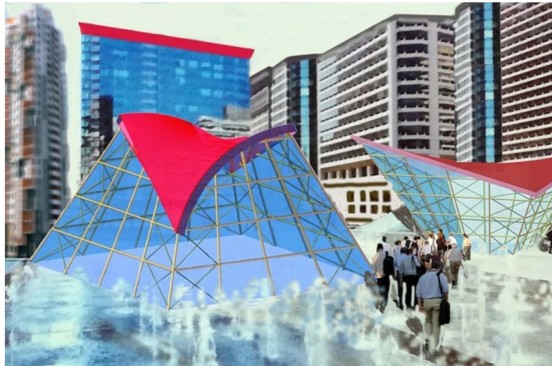

**Figure 29.** Two free-form buildings roofed with transformed shells located in a built environment.

The free-form buildings under consideration are visually attractive owing to the suitable shell shapes of roofs coherent with the oblique elevations. It is also noticeable how the color and regularity of elevation pattern affect the attractiveness and harmonious incorporation of the parametric free-form buildings into the built environment. Abramczyk presented a way of adopting fine proportions

between the parameters, leading to attractive building free forms [14]. The proposed method requires the designer to possess a certain predisposition to logical and spatial reasoning in the field of shaping free forms, as well as their texture, color, light, and shadow. Obtaining the required effect is conditioned by the individual artistic predispositions of the designer. The method assists the designers in managing their ambitious artistic goals.

## 9. Conclusions

The novel parametric description of the unconventional building free forms roofed with nominally plane-folded steel sheets transformed into shell sheeting is presented. The algorithm of the innovative method proposed for shaping the aforementioned forms with transformed corrugated shell roofs is based on the description. The method must be supported by the authors' application, which was created in the Rhino/Grasshopper program useful in parametric design

The proposed parametric description and the algorithm based on the description are related to the multidimensional aspects of the architectural free-form design. The use of the above parametric description is presented in detail regarding the example of shaping one relatively simple architectural free form with a transformed shell roof. The visualization of this form, which is shown in the last figure, is the effect of using the above description as well as the computer-aided method based on this description.

The article presents three basic types of the discussed architectural forms. Their general forms have diversified shapes, where their widths change at the height from the building base to the eaves in various ways. One of the presented forms expands along the height of the oblique façade walls from the base to the eaves (Figure 12), while the other contracts (Figure 13). In the third type of the presented forms (Figure 10), the width of the whole shaped building measured between the two opposite façade walls increases, while the width between the other pair of the opposite façade walls decreases in the vertical direction from the base to the eaves. The possibility of determining the various types of architectural forms, in which the elevation edges and planes are inclined to the vertical to various degrees, and roof surface rulings and eaves edges are inclined to the horizontal base plane, is clearly demonstrated. This proves the sensitivity of the proposed method to the harmonization of these forms with the built environments.

The intended effect consisting of creating an attractive unconventional architectural form should be achieved not so much by adopting the values of the proposed parameters, as much as by adopting the proportion between these values and one basic parameter, which is called the reference one. In the case of the chosen architectural form, the reference parameter is $ptr_{13} = 20{,}000$ mm, which describes the width of the architectural form. The adopted ratios are $ptr_1/ptr_{13} = ptr_2/ptr_{13} = 1.17$, $ptr_i/ptr_{13}$ for $i$ = three to six, $ptr_j/ptr_{13} = 0.64$ for $j$ = seven to 10, $ptr_{11}/ptr_{13} = 0.090$, $ptr_{12}/ptr_{13} = 0.036$, $ptr_{14}/ptr_{13} = 0.036$, $ptr_{15}/ptr_{13} = 0.025$, and $ptr_{16}/ptr_{13} = 0.15$. The adopted proportions and values make it possible to define the roof and elevation lines, including roof directrices. On the basis of the shape and mutual position of the directrices, the supporting conditions are calculated, and followed by the smooth shell models of the subsequent folds of the roof shell.

The authors' computer application supports these calculations. The application contains two basic conditions determining whether the created simplified smooth models guarantee the effectiveness of the fold's transformations during the assembly of these folds into the calculated places arranged along roof directrices. The first condition concerns the equality of the surface areas of a smooth model of a fold before and after its transformation. The experimental tests and computer analyses have shown that for folds of different profiles, and therefore of different lateral stiffness, the above areas differ relatively little compared with the accuracy of shell modeling. Further detailed experimental tests in this field are necessary in order to develop a function correcting the surface area of each fold after transformation, depending on its lateral stiffness. The second condition concerns the location of the fold's contraction along its length, and has to be rigorously observed, because even a relatively small change in the position of this contraction in relation to the length of the fold results in a significant

change in the proportion between the lengths of both its supporting lines *e* and *f*, and this significantly affects the transverse stresses, which is decisive for the value of the fold's initial effort.

As the thickness of each shell roof should be conspicuous, that is, significant in relation to the height of the building, both its upper and lower surfaces can be made up of the transformed corrugated sheeting. Both should usually be determined by means of the method, despite really small differences in the curvature of these surfaces. This results from even small differences in the supporting conditions of the folds of both shell sheetings inducing additional changes in the supporting line length. After the lengths of these changed lines have been added up, major differences arise in both the length of the supporting line of the entire roof shell, as well as in the spacing of the fixing points of these folds along the roof directrices. Additionally, it should be taken into account that these folds are usually supported by additional intermediate directrices at their length.

The architectural form of any designed building should be internally consistent; this means that the shape, position, and orientation of its characteristic straight and curved edges, as well as the flat and curved surfaces of the roof and façade, must be integrated. This integration must be taken into account both at the step of general form specification and when shaping the façade pattern. For this purpose, the proposed description includes parameters that define the regular form of the elevation pattern. In the case of the architectural form selected for discussion, a simple, equal division of each façade wall into horizontal and vertical glass strips separated by lines obtained by dividing each of its four edges into sections of equal length has been used. As the authors' analysis of the division of glass elevation planes into uneven strips or pattern diagonal orientation—which affects the integrity of the architectural form and its sensitivity to the built environment—is not complete yet, its results are not presented in the article.

The authors intend to continue and extend their research to the following areas: (1) the parametric description of free forms of complete buildings and their structures roofed with transformed corrugated shells; (2) the search for rational structural systems dedicated to the buildings under consideration here; and (3) the development of numerical models calibrated on the basis of their experimental research and exhibiting the geometrical and mechanical properties of elastically transformed thin-walled folded shells.

In the first case, the authors propose to analyze the possibilities of joining a few individual free forms of all three previously described types into a single structure with folded or segmented elevations and roof, which can be even more sensitive to the built environments than some complete forms. In the second case, due to the oblique orientation of the edges and surfaces of the façade and roof, it is necessary to adjust the shape of the structural system not only to the shape of the architectural form and its elements, but also to the character and direction of the characteristic load. Building construction has to guarantee an appropriate stiffness of architectural form, especially along the oblique edges of the roof and façade. The authors intend to conduct the analyses of various single-branch and multi-branch forms of structural elements such as poles and roof girders, depending on the architectural form dimensions and the roof span.

**Author Contributions:** J.A. carried out research and analyses, visualized and interpreted the results, created models and method as well as wrote all sections of the paper. J.A. was the supervisor and the project administrator. A.P. participated in the concept and investigations as well as funding acquisition.

**Funding:** The resources of the Rzeszow University of Technology.

**Conflicts of Interest:** The authors declare no conflict of interest.

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
