# Peer review of "Responsive Parametric Building Free Forms Determined by Their Elastically Transformed Steel Shell Roofs"

_buildings, doi:10.3390/buildings9020046_

Round 1
Reviewer 1 Report
This study is to present a method of shaping free form buildings of innovative general architectural forms resulting from the geometrical and mechanical properties of nominally, and their shell roofs in particular. This addressed an practical topic, but my major concern is that the novelty is unclear and some new finding should be discussed and proposed clearly in conclusion.
Some of my specific comments are listed below.
Line 78: some symbol should be clearly defined, such as symbols “A”, “B”, and “C” in Figure 3.
Line 125: The authors mention that “The method harmonizes and optimizes the static-strength ….”. How to verify that the designed results are optimized and it should be stated in conclusions.
Line 283: The novelty new finding should be discussed and proposed clearly in conclusion.
Author Response
One of the authors made the improvement of the submitted and revised manuscript. The co-author has approved the improvement of the resubmitted manuscript.
The content and figures included in the improved manuscript and the manner they are presented were modified or significantly changed in each of the sections. I would have to mark almost all parts in red, therefore I did not do that.
Thank you for the sets of critical reviews and comments for technical improvement of this manuscript. My response is as follows. I think that I take account of all suggestions. In particular, I described in details the process and conditions related to the optimization of the fold’s shape. The novelty of the achieved results is discussed. I defined the indicated symbols. Finally, the conclusions were improved and widely discussed.
Reviewer 2 Report
Find the attached document, which is the review for the Authors.

Author Response
One of the authors made the improvement of the submitted and revised manuscript. The co-author has approved the improvement of the resubmitted manuscript.
The content and figures included in the improved manuscript and the manner they are presented were modified or significantly changed in each of the sections. I would have to mark almost all parts in red, therefore I did not do that.
Thank you for the sets of critical reviews and comments for technical improvement of this manuscript. The summary of my specific response is as follows. I try to take account of all suggestions. It was not easy but very helpful in the new presentation of the content.
Following the reviewer’s suggestions, I changed the logic of the presented content of the manuscript. The introduction and literature are completed. I redefined the aims and discussed in details the solutions to the problems arising during the implementation of the aims. I also modified the content of the abstract significantly. Most of the text and figures in almost all sections are changed or added to explain the methodology used. The main modification consists in clarifying the objectives of the article, and discussing the proposed problems and methods of solving them. I changed the approach to the presentation of the main topics and conclusions of the article. The detailed discussion on the obtained results is added. I added Section 8 presenting a qualitative description of one selected architectural free form, on which example the innovative parametric description of the discussed building free forms is presented. I presented the role of a computer program written by me. The possibilities of further research and analysis are also discussed.
Round 2
Reviewer 2 Report
The Authors have performed an excellent job. They considered how I had commented their article and suitably addressed all my comments. Not only did they carefully and correctly addressed the issues raised in my review, but also they blended my suggestions or criticisms and further personal developments or contributions, which have further enriched the article.
Now, the article really adds to the subject and the presentation saves the readers’ effort to understand the message that the article aims at conveying.
Thus, I strongly recommend that the revised version of the article that has been resubmitted is accepted and published in the present form.